# Prevalence and Misreporting of Illicit Drug Use among Electronic Dance Music Festivals Attendees: A Comparative Study between Sweden and Belgium

**DOI:** 10.3390/toxics12090635

**Published:** 2024-08-29

**Authors:** Kristin Feltmann, Bert Hauspie, Nicky Dirkx, Tobias H. Elgán, Olof Beck, Tina Van Havere, Johanna Gripenberg

**Affiliations:** 1STAD, (Stockholm Prevents Alcohol and Drug Problems), 11364 Region Stockholm, Sweden; tobias.elgan@ki.se (T.H.E.); johanna.gripenberg@ki.se (J.G.); 2Centre for Psychiatry Research, Department of Clinical Neuroscience, Karolinska Institute & Stockholm Health Care Services, 11364 Region Stockholm, Sweden; olof.beck@ki.se; 3Belgium Substance Use and Psychosocial Risk Behaviours (SUPRB), HOGENT University of Applied Sciences and Arts, 9000 Ghent, Belgium; bert.hauspie@hogent.be (B.H.); nicky.dirkx@hogent.be (N.D.); tina.vanhavere@vandenbroucke.fed.be (T.V.H.)

**Keywords:** drug testing, drug consumption, breath sampling device, drinking, intoxication, biological sampling

## Abstract

Illicit drug use is common among attendees of electronic dance music (EDM) festivals, but is often significantly underreported by participants. The current study aimed to compare the prevalence and over- and under-reporting of illicit drug use among attendees at EDM festivals in two European countries with distinct drug laws and cultures. Self-reported data regarding recent drug use were collected through interviews. Participants’ blood alcohol concentrations were measured using a breathalyzer. Recent illicit drug use was assessed through sampling microparticles in the breath and consequent off-site analysis through liquid chromatography and tandem mass spectroscopy. Illicit drug use was higher in Belgium than in Sweden as indicated by self-reports (56.8 vs. 4.3%) and drug testing (37.2 vs. 12.5%). Underreporting was higher in Sweden than in Belgium; in Sweden, only 2.6% reported taking an illicit drug other than cannabis, whereas 11.6% tested positive, while the corresponding figures in Belgium were 36.5% and 36.9%. In both countries, results from self-reporting and drug testing for specific drugs matched poorly at the individual level, indicating unwitting consumption of substances. This study indicates that the drug use prevalence and the likelihood of disclosure may differ between countries or cultures, which should be considered when choosing methods to investigate drug use prevalence.

## 1. Introduction

Music festivals constitute a setting in which young people meet and socialize. The great majority of attendees also consume alcohol during or prior to entering the festival area [1,2,3,4]. High levels of alcohol intoxication increase the risk of injuries, sexual risk-taking and various forms of violence, including sexual violence [5,6,7,8,9,10,11,12]. Furthermore, the use of illicit drugs is prevalent in nightlife settings [13,14,15,16,17,18,19], including music festivals [2,3,20,21,22].

Illicit drugs are potent psychoactive compounds, such as cannabis (THC: delta-9-tetrahydrocannabinol), cocaine, ecstasy (MDMA: 3,4-methylenedioxymethamphetamine), amphetamine, and hallucinogens. Each substance is associated with its own acute risk, including psychosis, hallucinations, panic attacks, myocardial infarctions, overheating, and violence [23,24,25,26,27,28]. Moreover, when these substances are taken together with alcohol or when various illicit drugs are taken within a given time window (polydrug use), the risk of negative consequences increases [3,9,10,13,14,17,21,29,30,31]. Attendees at nightlife events consume these substances to a higher degree than the general population, although prevalences varied between studies conducted in different countries [13,14,15,16,32,33,34,35,36,37]. However, little is known about the extent to which attendees of electronic dance music (EDM) festivals consume illicit drugs and whether the levels of consumption differ between countries.

Although drug use among nightlife attendees has been widely studied, few have measured the use of these substances at EDM festivals using bioanalytical tests [3,20,21]. For example, a study of a large 40 h EDM event on a cruise ship departing from Sweden demonstrated that 4% of attendees self-reported recent drug use, but 10% tested positive for an illicit drug, indicating underreporting of drug use [3]. Similarly, a study of six festivals in Norway showed that 6% self-reported drug use, but 11% tested positive for drug use [21]. Thus, these two studies indicate that biological testing is needed to estimate the prevalence of drug use among festival attendees, a notion which was previously derived from studies conducted in clubs in Norway and the US [15,17,33]. The underreporting of EDM events in clubs and festivals in Sweden was confirmed in our recent study [38].

Potential reasons for underreporting drug use may be recall or social desirability biases [39,40,41]. While recall bias concerns difficulties among respondents in recalling previous events and experiences, social desirability bias refers to respondents’ tendency to modify their answers to conform to social norms [39,40]. For instance, the respondent may overreport what is considered good behavior to maintain a good self-image or underreport what is considered bad behavior to avoid shame and embarrassment in front of the interviewer or bystander [40]. Moreover, if confidentiality is uncertain, respondents may fear legal sanctions or risk to their employment or reputation [39].

Furthermore, studies among American homes and veterans conducted in 2001–2002 and 1996, respectively, found that cannabis use was overreported, whereas cocaine use was underreported, indicating that social acceptance of the substance also influenced the likelihood of self-reporting [42,43]. Nevertheless, in a Swedish cruise ship study conducted in 2011, somewhat higher prevalence rates for cannabis and cocaine were obtained through self-reporting compared to drug testing, whereas amphetamine and ecstasy were clearly underreported. This could also indicate that participants were unaware of the actual substance they had consumed; for example, they had received amphetamine instead of cocaine. Furthermore, in a Norwegian festival study conducted in 2016, cannabis, ecstasy, and cocaine use was underreported. Hence, it is not fully understood whether underreporting is country- or substance-specific. 

Sweden and Belgium have distinct drug laws. While drug use is a criminal offence in Sweden, leading to fines or imprisonment (Act on Penal Law on Narcotics (1968:64)), Belgium has no laws against drug use. However, possession of drugs, including cannabis, is illegal in both countries (Belgium: Narcotic Drug Act of 24 February 1921, Sweden: Act on Penal Law on Narcotics (1968:64)). In Belgium, cannabis possession for personal use can lead to fines or up to one year of imprisonment depending on the frequency of offenses within a certain period as well as the level of public disorder, but prosecution of cannabis possession for personal use should be given the lowest priority according to ministerial directives [44]. Differences in laws, policies, or norms may influence the extent of drug use as well as the over- or under-reporting of drug use.

Therefore, the present study’s aim was to compare the prevalence, as well as over- and under-reporting of illicit drug use among attendees at music festivals in Belgium and Sweden.

## 2. Materials and Methods

### 2.1. Settings

This study was conducted at three EDM festivals: a large two-day festival in Sweden, and two smaller festivals in Belgium. Data were collected during the summer of 2018 (June to September). Festival organizers were contacted ahead of time, informed about the study, and approved the data collection.

### 2.2. Procedure

The procedure was similar to that described previously [1,3,17,45]. Briefly, cross-sectional data were collected by a team of four to seven trained research assistants. Two teams were collecting data simultaneously at different locations at the larger Swedish festival. Each team consisted of several interviewers and one recruiter who invited festival attendees to participate in the study. Prior training of research assistants included how to approach participants, informed consent (voluntary and anonymous participation), the interview process, mouth rinsing before breath measurements, and how to correctly handle the breathalyzer and drug breath sampling device. Recruiters invited every third person to pass through an imaginary line to participate in the study [3,32,45]. To minimize refusal, individuals accompanying the person approached were also invited to participate [46]. Upon refusal, the estimated age and gender of the approached person were recorded.

First, the participant was given a cup of water for mouth rinsing (necessary for correctly measuring alcohol and drugs in the exhaled breath). Participants were then asked questions on demographic information (age, gender and employment status) as well as smoking and alcohol drinking habits using the AUDIT-C (Alcohol Use Disorder Identification Test-C) [47,48]. Blood alcohol concentration (BAC) was assessed using an alcohol breathalyzer (Dräger Alcotest 6820). Participants were asked to complete a questionnaire on their own use of different types of illicit drugs. At the end of the interview, the participant was asked to exhale ten times—breathing normally—through a device containing a filter collecting aerosol microparticles (BreathExplor^®^, Munkplast, Uppsala, Sweden) [49]. The interview and breath sampling were completed in about ten to fifteen minutes. Stickers containing identical barcodes were attached to the questionnaire and device. Devices were stored in cooler boxes (+4 °C) during data collection and then transferred to freezers (−20 °C) and stored until analysis. Finally, a small incentive was offered to the participants (chips, chocolate, and chewing gum). When invited to the study, participants were informed that it focused on alcohol and drugs, and were made aware of the offered incentive, but were not told about the drug testing to avoid influencing the accuracy of their self-reporting.

### 2.3. Chemical Analysis

The filter was removed from the device, and microparticles were extracted using a methanol solution. Samples were analyzed as previously described [49,50] using a combination of liquid chromatography and tandem mass spectroscopy, including electrospray ionization and selective reaction monitoring. This method has high sensitivity and specificity, with a detection limit of one picogram. Each sample was analyzed for 47 different substances (Table 1), including illicit drugs and their metabolites.

The maximum time window of detection for the analyzed compounds was between 24–48 h after consumption, except for delta-9-tetrahydrocannabinol (THC), for which the detection window was up to six hours [51]. Positive test results for metabolites were treated as an indication for illicit drugs (see Table 1): ritalinic acid for methylphenidate; EDDP for methadone; norbuprenorphine for buprenorphine; benzolyecgonine for cocaine; and morphine, codeine, 6-acteylmorphine and 6-acetylcodeine for heroin [52].

### 2.4. Statistical Analysis

Data were analyzed using SPSS (version 27). Chi2-analysis was conducted to compare demographic factors, as well as smoking, BAC levels, AUDIT-C score category, and self-reported and measured drug use prevalence between countries. A t-test was used to compare BAC levels between countries. Binomial regressions were conducted to investigate if country-specific differences in self-reported and tested drug use remained after controlling for age, gender, education status and frequency of going out to clubs.

The level of agreement between self-reported data and biological test results were measured with Cohen’s kappa (k) and defined as follows: no agreement < 0.20, minimal 0.21–0.39, weak 0.40–0.59, moderate 0.60–0.79, strong 0.80–0.90 and almost perfect > 0.90 [53]. The significance level was set at 0.05.

### 2.5. Ethical Considerations

This study was conducted in accordance with the Declaration of Helsinki and approved by the Regional Ethical Review Board in Stockholm (2017/1207-32) and the Ethics Committee of the Faculty of Psychology and Educational Sciences, Ghent University, Belgium (2017/14/Bert Hauspie, 14 April 2017). Informed consent was obtained from all participants involved in the study.

## 3. Results

### 3.1. Participants

In Sweden, 77% (*n* = 669) of the 864 individuals invited agreed to participate. Participants’ median (IQR) age was 21 (20–24) years, and 55% were men. Among those who did not participate, the median (IQR) age was 22 (20–25) years, and 71% were men. In Belgium, 88% (*n* = 329) of the 376 people invited agreed to participate. Participants’ median (IQR) age was 21 (19–25) years, and 71% (*n* = 235) were men. Among those who did not participate, the median (IQR) age was 23 (20–28), and 62% were men.

A comparison of demographic data revealed that in Belgium, compared to Sweden, a larger proportion were men, more participants were students, fewer were working full-time, and fewer were going out to clubs at least six times a year (Table 2). Comparing substance use between countries demonstrated that, in Belgium compared to Sweden, a larger proportion of the participants were daily smokers, hazardous drinkers, and had taken illicit drugs during their lifetime (Table 2). However, the measured BAC level was higher in Sweden than Belgium (mean ± SD: 0.079 ± 0.056% vs. 0.051 ± 0.054%, t(996) = 7.74, *p* < 0.001).

### 3.2. Recent Use of Illicit Drugs: Self-Reporting and Biological Testing

In Sweden, drug testing revealed a higher prevalence of illicit drug use (any substance) as well as cocaine, ecstasy, and amphetamine use than self-reporting (Table 3). In Belgium, self-reporting resulted in a higher prevalence than drug testing, mainly regarding cannabis and ecstasy but not amphetamine use. In both countries, cannabis was detected in only a few samples, likely because of the short detection window of up to 6 h. With the exception of hallucinogens and heroin, self-reported recent drug use was significantly higher in the Belgian sample than in the Swedish sample (Table 3). Furthermore, a significantly larger proportion of participants in Belgium than in Sweden tested positive for cocaine, amphetamine, ketamine, and heroin.

Regarding self-reported recent drug use, the binomial regression model was statistically significant overall (χ^2^(9) = 385.84, *p* < 0.001, Nagelkerke R^2^ = 0.511). The likelihood of reporting recent drug use was significantly higher among participants from Belgium compared to Sweden (OR = 35.79, *p* < 0.001), males compared to females (OR = 2.28, *p* < 0.001), and those who regularly go to clubs (OR = 1.83, *p* = 0.022). Conversely, this probability was significantly lower among high school students compared to full-time employees (OR = 0.281, *p* = 0.001) and was not significantly influenced by age (*p* = 0.065).

Similarly, the binomial regression model for testing positive for at least one illicit drug was also statistically significant overall (χ^2^(9) = 94.93, *p* < 0.001, Nagelkerke R^2^ = 0.149). The probability of a positive drug test result was significantly higher in Belgium compared to Sweden (OR = 4.76, *p* < 0.001) and among those who regularly go to clubs (OR = 1.62, *p* = 0.040). However, it was not significantly affected by age (*p* = 0.119), gender (*p* = 0.400), or employment status (*p* = 0.099).

Out of 34 individuals who tested positive for heroin, 33 tested positive for 6-acetylmorphine (of which eight were also positive for 6-acetylcodeine) and one tested positive for codeine but not for other heroin metabolites or dihydrocodeine.

Apart from the drugs listed in Table 3, the following substances were also found in the samples: tramadol (n = 2 in Belgium, n = 1 in Sweden), oxycodone (n = 1, Sweden), oxazepam (n = 1, Sweden), temazepam (n = 2, Sweden), methamphetamine (n = 2 in Belgium, n = 1 in Sweden), MDA (n = 1 in Sweden), mephedrone (n = 1, Belgium), pregabalin (n = 2, Sweden), gabapentin (n = 1 in Belgium, n = 1 in Sweden), and methylphenidate or metabolite (n = 6, Sweden).

In Belgium, no differences existed between genders, as 37% of both men and women tested positive for an illicit drug (χ^2^(1) = 0.00, *p* = 0.993). In Sweden, however, there was a difference, as 14% of men and 11% of women tested positive (χ^2^(1) = 1.91, *p* = 0.166).

To further investigate over- and under-reporting, the results of drug testing and self-reports were matched at the individual level. Among participants who had not self-reported the recent use of any illicit drug, approximately 11% in Sweden and 28% in Belgium tested positive for at least one illicit drug (Table 4), indicating underreporting. In both countries, there was no agreement between self-reported and measured use of cannabis and only minimal agreement regarding cocaine use. In Belgium, there was no agreement between ecstasy and minimal agreement for amphetamines. In Sweden, there was minimal agreement on ecstasy and no agreement on amphetamines. Almost none of the participants who had reported cannabis use tested positive, potentially due to the shorter detection window of up to six hours.

## 4. Discussion

Drug testing revealed that recent drug use prevalence was higher among attendees at the EDM festivals investigated in Belgium than among those in Sweden. Furthermore, participants in Sweden, but not in Belgium, underreported their recent drug use. Nevertheless, participants in Belgium often reported having taken substances other than those that tested positive. Drug testing seems to measure recent drug use prevalence more accurately in Sweden, in contrast to Belgium, where recent drug use was more honestly reported.

### 4.1. Drug Use Prevalence Differences

In Sweden, the prevalence of illicit drug use as discovered by drug testing was similar to that reported in studies on EDM events and festivals in Northern Europe [3,21]. In Belgium, the prevalence of drug use was higher than that in Sweden for both illicit drugs overall and for several illicit drugs separately. The higher prevalence of drug use in Belgium aligns with the broader European trends. A report by the European Union Drug Agency (EUDA, formerly known as EMCDDA) positioned Belgium as a significant hub for substances such as cocaine, amphetamine, and MDMA [54]. This central role in the drug trade, and thereby high availability, could contribute to the higher drug use prevalence observed at the Belgian festival scene. In contrast, Sweden’s lower prevalence and higher under-reporting rates may reflect its stricter drug policies and the social stigma associated with drug use. The wastewater analysis results from 2018 and 2019 presented on the EUDA webpage supported these findings, showing varying levels of drug use across different cities and substances [55]. Whereas cocaine and MDMA were found at higher levels in Belgian cities than in Stockholm, amphetamine was found at higher levels in Stockholm than in Belgian cities such as Antwerp or Oostende, but at much lower levels than in Brussels. Nevertheless, differences in the type of festival, gender, and employment status (see Section 4.4), as well as the fact that drug use is illegal in Sweden but not in Belgium, could also contribute to differences in drug use prevalence.

### 4.2. Self-Reports vs. Drug Testing

In Sweden, the drug use prevalence was higher when using drug testing than when using self-reporting, while in Belgium, levels were similar once cannabis was excluded (owing to the shorter detection window). In the current study, drug use was underreported in Sweden by a factor of four, which is similar to studies that reported factors of two to three [3,21]. Using either self-reporting or drug testing, the prevalence of drug use for most drugs was higher in Belgium than in Sweden, with the exception of heroin, which was not self-reported. Heroin use could have been unknown to the participants or potentially associated with higher stigma and therefore, less likely to be declared in self-reporting. However, when matching drug testing and self-reports at the individual level, agreement was minimal for most drugs, and both under- and over-reporting were found. These results indicate that people are not aware of the substance they consume, signaling unwitting drug consumption; therefore, drug testing is superior to self-reporting in identifying substance use and determining its prevalence. Furthermore, drug checking services could be a relevant harm reduction initiative at EDM events or festivals [56,57,58] but are also characterized by challenges regarding resources, logistics, and scientific evidence of efficiency [59]. Unwitting drug consumption can result from adulteration, mislabeling, or the sale of counterfeit drugs in illegal, unregulated markets.

### 4.3. Potential Reasons for Over- and Underreporting

There could be a variety of reasons for over- and under-reporting, both common and different, across countries. Owing to stricter laws and their increased enforcement in Sweden compared to Belgium, underreporting could be related to fear of legal consequences, as well as social stigma. Studies on drug use [15,45] found similar trends in underreporting due to social desirability bias, and legal concerns have been reported. In Belgium, a more permissive attitude toward drug use and less stringent policies may contribute to more honest reporting. Similar to questions regarding drug use, other surveys on sensitive topics have demonstrated social desirability bias (see review [40]). Furthermore, the settings and procedures can influence the likelihood of reporting drug use. To ensure comparability, researchers from both countries assisted in data collection and ensured that recruitment, interviews, sampling, and sample handling were similarly conducted. The fact that the survey was anonymous, no identifying information was recorded, and the test results could not be seen on-site may have limited the extent of underreporting and dropping out of the study. 

Another reason for underreporting or overreporting could be that, with increasing intoxication levels, there may be a negative effect on memory. Although more people in Belgium than Sweden were identified as hazardous drinkers, BACs were higher in Sweden than in Belgium. Nevertheless, illicit drug use is higher in Belgium, which may have led to higher levels of drug intoxication. These findings could possibly be explained by legal differences between the two countries, i.e., the legal drinking age is 16 in Belgium and 18 in Sweden, and that drug use is not criminalized in Belgium. However, studies suggest that while alcohol intoxication decreases the number of details being recalled (completeness), the accuracy and reliability of recall memory are not compromised [60,61]. Moreover, alcohol intoxication did not seem to increase honesty or impede people’s ability to conceal information [61,62]. Furthermore, the regression model indicated that men are more likely to report recent use of illicit drugs compared to women. This finding is consistent with our previous study, which showed that although fewer women than men tested positive for drugs, those women who did test positive were less likely to report drug use, suggesting a tendency toward underreporting among women [38].

### 4.4. Limitations

This study’s limitations include a potential gender imbalance in the participant pool, although even among those refusing there was also a male majority. In addition, our previous studies suggested a similar male majority in frequent EDM event attendees [37,63,64]. Similarly, there were differences in employment statuses between countries, which could influence drug use patterns and reporting, but this could have also been related to the different types of events. The study’s findings are limited to specific festival settings in Belgium and Sweden and may not be generalizable to other contexts or countries. The Belgian festivals visited were smaller than the Swedish festivals and might have represented a more niched subculture in which illicit drug use was more common. During the summer before our study, a small EDM festival planned to be held at an old factory and with already sold tickets was not allowed to be held after a decision from the police three weeks before the event based on safety concerns owing to the building in combination with expected alcohol and drug intoxication. Therefore, we had trouble finding a similar small-niched festival in the year of our study, and although the one we chose was larger, it clearly played EDM. Furthermore, open-ended questions regarding the substances used were not included in the questionnaire, which could have potentially provided further insight into which substances the participants believed they had consumed, especially when slang or street names might have been more familiar to them.

## 5. Conclusions

This study provides crucial insights into the patterns of illicit drug use and reporting during music festivals in Belgium and Sweden. The observed differences underscore the need for culturally and legally sensitive approaches to address drug use in nighttime settings. These findings emphasize the importance of considering local drug policies, cultural attitudes, and reliable methods for measuring drug use in research and policymaking. In addition, our findings also allow law enforcement and medical personnel to better prepare for and respond to substance use-related incidents. Future research should explore in depth the cultural and legal factors influencing drug use reporting and develop and test strategies tailored to specific environments and cultures. Understanding these factors is crucial for developing effective drug prevention and harm reduction strategies, such as drug-checking services and how to communicate to those under the influence, which are sensitive to the specific cultural and legal contexts of different countries.

## Figures and Tables

**Table 1 toxics-12-00635-t001:** List of substances analyzed in the breath samples.

Cannabis (THC)	**Central stimulants**
Ketamine	Cocaine
**Opioids**	Benzoylecgonine (Cocaine metabolite)
*heroin metabolites*	Amphetamine
6-Acetylmorphine	Methamphetamine
Morphine	Ecstasy (MDMA)
6-Acetylcodeine	Mephedrone (4-MMC)
Codeine	MDA
*pain medications*	MDPV
Dihydrocodeine	4-Methylcathinone
Hydromorphone	Methylone (MDMC)
Tramadol	Butylone
O-Desmethyltramadol	PMMA
Oxycodone	*ADHD medications*
*opioid use disorder medications*	Ritalinic acid (Methylphenidate metabolite)
Methadone	
EDDP (methadone metabolite)	**New Psychoactive Substances (NPS)**
Buprenorphine	Alpha-PVP
Norbuprenorphine (Buprenorphine metabolite)	Pentedrone MDEA
**Benzodiazepines**	MBDB
Diazepam	BDB (MBDB metabolite)
Nordazepam	**Other medications (classed as narcotics)**
Oxazepam	Zopiclone
Temazepam	Zolpidem
Flunitrazepam	Pregabalin
Alprazolam	Gabapentin
Bromazepam	
Midazolam	
Lorazepam	

Alpha-PVP: alpha-pyrrolidinovalerophenone, BDB: 1,3-benzodioxolylbutanamine, EDDP: 2-ethylidene-1,5-dimethyl-3,3-diphenylpyrrolidine, MBDB: 1,3-benzodioxolyl-N-methylbutanamine, MDA: 3,4-methylenedioxyamphetamine, MDEA: 3,4-methylenedioxy-N-ethylamphetamine, MDMA: 3,4-methylenedioxymeth-amphetamine, MDMC: 3,4-methylenedioxy-*N*-methylcathinone, MDPV: methylenedioxypyrovalerone, 4-MMC: 4-methyl methcathinone, PMMA: para-methoxyamphetamine, THC: delta-9-tetrahydrocannabinol.

**Table 2 toxics-12-00635-t002:** Demographic data and substance use during lifetime.

	Sweden % (n)	Belgium % (n)	Chi-Square (df), *p*-Value
**Gender** Men Women Other	55.0 (366) 44.6 (297) 0.5 (3)	71.4 (235) 28.6 (94) 0.0 (0)	25.76 (2), <0.001
**Employment status** Full-time Part-time University student High-school student Unemployed Other	61.1 (409) 13.2 (88) 16.6 (111) 5.1 (34) 1.8 (12) 2.2 (15)	44.3 (143) 3.4 (11) 31.6 (102) 17.0 (55) 3.1 (10) 0.6 (2)	94.32 (5), <0.001
**Number of club visits** **per year** <6 times ≥6 times	15.0 (98) 85.0 (554)	22.5 (72) 77.5 (248)	8.30 (1), 0.004
**Smoking** Non-smoker Non-daily Daily smoker	53.6 (358) 32.8 (219) 13.6 (91)	32.1 (104) 16.4 (53) 51.5 (324)	163.74 (2), <0.001
**Drinking habits last year** Abstainers Non-hazardous Hazardous drinkers	3.4 (23) 40.7 (272) 55.8 (668)	2.7 (9) 29.2 (96) 68.1 (224)	13.82 (2), <0.001
**Last use of illicit drug** Never Several years ago During the last year	54.0 (361) 16.3 (109) 29.6 (198)	9.5 (31) 8.5 (28) 82.0 (269)	249.50, <0.001
**Ever-use of substances** Cannabis Cocaine Ecstasy Amphetamine Mushrooms LSD NPS Ketamine Heroin Benzodiazepines Opioid medication ADHD medication Anabolic androgenic steroids	42.6 (284) 20.3 (135) 18.2 (121) 9.2 (61) 7.2 (48) 5.3 (35) 3.5 (23) 2.1 (14) 0.3 (2) 11.4 (75) 9.1 (60) 9.9 (65) 1.5 (10)	84.6 (270) 53.7 (175) 63.9 (209) 34.4 (111) 34.8 (114) 28.0 (92) 17.2 (56) 45.1 (147) 10.4 (34) 20.2 (66) 19.9 (65) 19.8 (64) 1.2 (4)	

Participants: n = 669 (Sweden), n = 329 (Belgium). Data are missing for gender (n = 3), employment status (n = 6), club visits (n = 26), smoking (n = 6), drinking habits (n = 1), illicit drug use (n = 2), cannabis (n = 12), cocaine (n = 7), ecstasy (n = 5), amphetamine (n = 10), mushrooms (n = 5), LSD (n = 6), NPS (n = 8), ketamine (n = 8), heroin (n = 5), benzodiazepines (n = 16), opioid medication (n = 11), ADHD medication (n = 16), anabolic androgenic steroids (n = 13).

**Table 3 toxics-12-00635-t003:** Prevalence of recent illicit drug use obtained by drug tests using exhaled breath samples and self-reports during the past 48 h.

	Sweden		Belgium	
	Self-Reported Use Last 48 h % (n)	Positive Breath Test % (n)	Self-Reported Use Last 48 h % (n), *p*-Value ^5^	Positive Breath Test % (n), *p*-Value ^5^
Any illicit drug ^1^	4.3 (29)	12.5 (83)	56.8 (187), <0.001	37.2 (122), <0.001
Illicit drug other than cannabis	2.6 (17)	11.6 (77)	36.5 (120), <0.001	36.9 (121), <0.001
Cannabis/THC ^2^	2.4 (16)	1.1 (7)	46.3 (150), <0.001	0.3 (1), 0.215
Cocaine	1.7 (11)	5.1 (34)	15.0 (48), <0.001	11.9 (39), <0.001
Ecstasy	1.1 (7)	3.2 (21)	25.8 (82), <0.001	0.6 (2), 0.012
Amphetamine	0.2 (1)	3.2 (21)	5.9 (19), <0.001	16.2 (53), <0.001
Mushrooms ^3^	0.2 (1)	-	0.6 (2), 0.211	-
LSD ^3^	0.3 (2)	-	0.9 (3), 0.195	-
NPS ^4^	0.3 (2)	0.0 (0)	1.9 (6), 0.010	0.0 (0)
Ketamine	0.2 (1)	0.3 (2)	12.2 (39), <0.001	13.7 (45), <0.001
Heroin	0.2 (1)	0.3 (2)	0.0 (0), 0.486	10.4 (34), <0.001

Participants completed a self-reported questionnaire regarding their drug use during the last 48 h (self-reported). Breath samples were then analyzed for the presence of the same illicit drugs. A total of 669 people were interviewed in Sweden and 329 in Belgium, of whom 665 and 328 were tested for drugs in Sweden and Belgium, respectively. ^1^ self-reported drug use of at least one substance listed in the table. Positive test results for at least one of the 47 substances tested. ^2^ Analyses of breath samples can only detect cannabis within the last 6 h. ^3^ Magic mushrooms and LSD were not tested in breath samples. ^4^ Of the four NPS tested, none were found in the breath samples. Self-reported data for cannabis (n = 20), cocaine (n = 27), ecstasy (n = 26), amphetamine (n = 15), mushrooms (n = 10), LSD (n = 8), NPS (n = 13), ketamine (n = 13), and heroin (n = 11) are missing. ^5^ Chi-square test of corresponding values between countries.

**Table 4 toxics-12-00635-t004:** Agreement between self-reported and measured drug use.

	Tested Positive Sweden % (n Positive /n Total)	Cohen’s Kappa	Tested Positive Belgium % (n Positive /n Total)	Cohen’s Kappa
**Any illicit drug** did not self-report use self-reported use	10.6 (67/635) 55.2 (16/29)	0.236	28.4 (40/141) 43.9 (82/187)	0.147
**Illicit drug other than cannabis** did not self-report use self-reported use	9.9 (64/646) 76.5 (13/17)	0.245	27.9 (58/208) 52.5 (63/120)	0.246
**Cannabis** did not self-report use self-reported use	1.1 (7/635) 0 (0/16)	-0.015	0 (0/173) 0.7 (1/150)	0.007
**Cocaine** did not self-report use self-reported use	3.8 (24/638) 72.7 (8/11)	0.356	7.7 (21/271) 35.4 (17/48)	0.303
**Ecstasy** did not self-report use self-reported use	2.2 (14/644) 71.4 (5/7)	0.375	0.4 (1/235) 1.2 (1/82)	0.012
**Amphetamine** did not self-report use self-reported use	3.2 (21/659) 0 (0/1)	−0.003	13.7 (41/300) 63.2 (12/19)	0.269

Participants (n = 669). Data are missing for any illicit drug (n = 6), illicit drug other than cannabis (n = 7), cannabis (n = 24), cocaine (n = 30), ecstasy (n = 30), amphetamine (n = 19).

## Data Availability

Data are available upon request. Requests can be sent to kristin.feltmann@ki.se, tobias.elgan@ki.se or johanna.gripenberg@ki.se.

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
