# Peer review of "Prevalence and Misreporting of Illicit Drug Use among Electronic Dance Music Festivals Attendees: A Comparative Study between Sweden and Belgium"

_toxics, 2024, doi:10.3390/toxics12090635_

Round 1

Reviewer 1 Report

Comments and Suggestions for Authors

Thank you for the opportunity to review this well-written paper describing differences in electronic dance music festival attendees in Sweden and Belgium. The paper makes a modest contribution to the literature but could be strengthen by addressing the relative minor issues outlined below.

1.      The abstract would benefit from some justification/rationale for the comparison between Sweden and Belgium. This is well outlined in the manuscript and represents the most interesting findings of the paper so it worth including in the abstract.

2.      In the introduction (page 2, lines59-75) potential reasons for under-reporting drug use are discussed, including social desirability bias. However, there is no mention of non-reporting due to perceived or potential punitive outcomes in drug criminalised environments. In other words, where substance use is criminalised, there is both a perceived and real risk of penalty if reported.

3.      Page 3, line 98 “…by a team of four-seven trained research assistants.” This sentence is unclear. I assume you mean “…four teams comprised of seven trained research assistants.”

4.      It is unclear exactly why the AUDIT-C, a measure of alcohol problems, is used. Whilst this provides some useful data, it is unclear why explicitly alcohol use problems (rather than just current alcohol consumption) would be examined especially when only current illicit drug use is assessed.  This warrants some explanation.

5.      Can you please provide an estimate of the time (e.g., mean minutes) it took participants to complete the study procedures.

6.      The “occupation” category might be better phrased as “employment status”.

7.      The first sentence of the discussion is well summarised and gets to the heart of the findings.

8.      It would be useful to consider what is known about drug markets and how the findings might be interpreted in that context. This is especially relevant when discussing self-reports vs drug testing (page 8, lines 247-254).

9.      Page 8, lines 245-246 – “Heroin use was either…less likely to be declared in self-reporting”. Although likely, it needs to be clear the statement regarding stigma is supposition.

10.  Line 271-272 – this is an interesting finding, but authors should consider whether this might be a feature (or product) of the legal status of alcohol vs illicit substances in Sweden and what can be inferred from that assumption. This would be an interesting addition but could still be kept brief.

11.  The gender imbalance (and ‘occupation’) is not necessarily a limitation of the study, it may be a finding. This can only be inferred if more is known about the demographic characteristics of EDM festival attendees in both countries. Perhaps men are over-represented in these groups? The exploratory nature of the study means that the limitation is the response rate or sampling strategy which may have led to the gender imbalance. However, without information about EDM attendees in both countries conclusions as to representation cannot be determined. The issue around similarity of festival type is relevant and may offer insight?

Reviewer 2 Report

Comments and Suggestions for Authors

The authors present an epidemiological survey investigating illicit-drug use among electronic dance music festivals attendees. In particular, they  compared the prevalence, over- and under-reporting of illicit drug use among  attendees at electronic dance music festivals in Belgium and Sweden. 

The study is scientifically sound, well-presented and of interest for the readers.

Minor points

lines 163-164 please add unit of measurement 

Table 2 please correct into "anabolic androgenic steroids", there is a typo

Comments on the Quality of English Language

The paper is well written. There is only a typo. Table 2 please correct into "anabolic androgenic steroids"

Reviewer 3 Report

Comments and Suggestions for Authors

Overall a very good presentation of data reflecting EDM festival drug use patterns in Belgium and Sweden. The use of two countries provided contrast and comparison. The authors do well to present the study, the data collection and results, and some challenges. The report is well written and interesting to read. There are few major concerns and some moderate to mild concerns which are listed below.

1) Were participants informed that they would have collection of biosamples for assay of various drugs before the questionnaire portion of data collection?  This should be specified (if not already) since it could bias the accuracy and upfront nature of the responses from the participants. That is, if participants expect their responses will be justified by lab testing, they might have answered differently than if they didn't have that expectation.

2) Throughout, tetrahydrocannabinol is identified, and presumably indicates "delta-9-tetrahydrocannabinol" (or d9-tetrahydrocannabinol). Given the increased use of delta-8-tetrahydocannabinol (not to mention synthetic cannabinoids), it would be best to specify which cannabinoid was studied here.

3) In Table 1, drugs/medications potentially detected by LC/MS are listed. Were other drugs/compounds available to list in the self-report?

4) Did self-report/interview data collection include options for other drugs not otherwise specified (e.g., synthetic cannabinoids)?

5) In tandem with above, were drugs identified by a variety of slang and street names to provide inclusiveness for identification?

6) Was any analysis of covariates with age, sex, or occupation possible or performed? Since there was a significant difference in these and other demographics between the groups it is a significant limitation in the interpretation of the results of the study, attributing the finding primarily on differences between Sweden and Belgium. If these correlations/covariates were not possible to be performed (due to how data was collected and coded), it should be explicitly stated as such and identified as a limitation of the study.

7) The authors could include a bit of projection and consideration about the value of the research included in this manuscript. For instance, this could be foundational for future research, including communicating to those using drugs about identifying specific substances (MDMA or opioid testing services), or designing specific drug intervention programs that are country/culturally specific.

Comments on the Quality of English Language

The first instance of MDMA in the introduction is not spelled out, though it is for THC, and later done so in Table 1. It should be defined in the first instance.

Introduction, 2nd paragraph: "THC-tetrahydrocannabinol" would be better with a colon than hyphen to indicate definition.

The in-text reference to EMCCDA webpage, "drug use penalties at a glance" should probably be more clearly identified as a citation to avoid confusion as a method used by the authors (ie, glancing at the penalties), by capitalization, quote marks, or italics. Referral to a style guide may be necessary.

2.2 Procedure, 2nd sentence: "four-seven" might be more clear as either "four to seven" or "4 - 7".

2.4: "no agreement 0.20" currently has the pound (currency) sign in front of 0.20, when it seems to require "<". Please clarify.

Table 2: "baily smoker" needs correction to "daily smoker"

and "anabola andondrongenic steroids" needs correction to anabolic androgenic steroids as noted in the legend.

Reviewer 4 Report

Comments and Suggestions for Authors

The article describes the prevalence, over-reporting, and under-reporting of illicit drug use among attendees of electronic dance music festivals in Sweden and Belgium. The topic is relatively original and the question is worth exploring.

The work may contribute to the advancement of current knowledge. It may be important in areas such as public health, law enforcement, and festival management to make the environment relatively safe for audiences attending dance music festivals through risk reduction.

Work is presented in a satisfactory manner.

Specific comments to the manuscript:

1.) Materials and Methods, page 3, line 128: The term "detection limit" is well defined as the lowest concentration of analyte that can be reliably detected and reflects the precision of the instrumental response obtained by the method when the concentration of analyte is zero. This term has nothing to do with the time window of detection of some illicit drugs in biological samples, as you can see. The sentence should be corrected as meaning.

2.) Results, page 5, line 175: "6-Acetylmorphine" should be written with a small initial letter.

3.) In Table 2 and Table 3, there are discrepant rows in some places. This needs to be corrected to properly understand the data presented.

4.) The contribution of the work should be discussed more in the conclusion. It should be noted that such studies inform public health interventions and policy decisions. For example, trends in drug use allow the formulation of harm reduction strategies, such as providing information on the effects of substances and how to use them safely. It also allows law enforcement and medical personnel to better prepare for and respond to incidents.

5.) The supplementary file contains a huge table of data that is completely meaningless and difficult to understand. This file looks like a file prepared for import into a statistical processing program, but not a file containing information readily available to the reader. I strongly recommend that this file either be refactored into an understandable form or removed.

Comments on the Quality of English Language

1.) Abstract, page 1, line 13: The term "illicit drug" can be used without dash.

2.) Abstract, page 1, line 25-26: The sentence "This study indicates that drug use prevalence and the likelihood of disclosing use can differ ..." would be better worded as "This study indicates that the drug use prevalence and the likelihood of disclosure may differ ...".

3.) page 2, line 63 and line 85; page 8, line 268 and line 270 and page 9, line 280: In some places the words "might" and "could" could be replaced with "may".

4.) Discussion, page 7, line 218 and line 219: Write "..., where recent drug use was more honestly reported." instead of "..., where recent drug use has been reported more honestly".

5.) Discussion, page 8, line 266 and line 267: Write "..., and sample handling were similarly conducted" instead of "...,  and sample handling were conducted similarly".
